# Influence of Zooplankton and Environmental Factors on Clear-Water Phase in Lake Paldang, South Korea

**DOI:** 10.3390/ijerph18137205

**Published:** 2021-07-05

**Authors:** Younbo Sim, Myeong Seop Byeon, Keonhee Kim, Soon Ju Yu, Jong Kwon Im

**Affiliations:** 1Han River Environment Research Center, National Institute of Environmental Research, 42, Dumulmeori-gil 68beon-gil, Yangseo-myeon, Yangpyeong-gun 12585, Korea; sumatra0@nate.com (Y.S.); zacco@korea.kr (M.S.B.); ysu1221@korea.kr (S.J.Y.); 2Department of Environmental Health Science, Konkuk University, Seoul 05029, Korea; 3Human and Eco-Care Center, Department of Environmental Health Science, Konkuk University, Seoul 05029, Korea; skyopera@konkuk.ac.kr

**Keywords:** water quality, phytoplankton, secchi depth, hydraulic retention time, Bayesian model, Cladocera

## Abstract

Lake Paldang is a complex water system with both fluvial and lacustrine characteristics and the largest artificial dam lake in South Korea. In this study, the different occurrence patterns and causes of the clear-water phase (CWP) were investigated using water quality and hydrological factors at four sites in Lake Paldang. Among the environmental and other factors associated with CWP occurrence, secchi depth and turbidity exhibited significant correlations with precipitation, hydraulic retention time (HRT), and individual and relative abundance (RA) of zooplankton (Cladocera) (*p* < 0.01). Hence, a change in the HRT because of precipitation could alter the CWP. The Cladocera individuals and RA showed significant correlations with the water temperature, indicating that the emergence of Cladocera in spring as the water temperature rises could cause the CWP. Sensitivity assessments conducted using Bayesian models demonstrated different CWP occurrence sensitivity relationships for the river-type, lake-type, and shallow and deep sites. Turbidity, secchi depth, and zooplankton factors also showed sensitive relationships with CWP occurrence for all sites. The sensitivity to precipitation and HRT was higher in the river-type sites. The lake-type sites, with common Cladocera emergence and long HRT, favored CWP occurrence. Thus, CWP occurrence was dependent on the site characteristics and climate conditions.

## 1. Introduction

Generally, the clear-water phase (CWP) primarily occurs in the succession pattern of phytoplankton in mesotrophic or eutrophic lakes during spring in temperate regions. According to Dröscher, et al. [1], the CWP is associated with the following conditions: (1) increased secchi depth, (2) a reduction in the phytoplankton biomass, and (3) an increase in the zooplankton biomass. This is also consistent with the results of Sommer et al. [2].

The emergence of phytoplankton in spring is caused by increased light, whereas zooplankton exhibit seasonal trends linked to increasing water temperature [3]. In particular, shallow and extensive lakes are characterized by high light transmission, except those heavily contaminated with natural or artificial organic matter, which increases the phytoplankton density [4,5,6,7]. The occurrence of the CWP is favored by an increase in fine phytoplankton and an increase in the biomass of Cladocera order, which are mostly filter feeders [8,9,10]. Furthermore, chlorophyll-*a* (Chl-*a*) and zooplankton show an inverse relationship, and phytoplankton biomass is the lowest when, among zooplankton, the biomass of Cladocera (*Daphnia*) is the highest [11]. Analysis of cumulative data of the past 20 years reveals a strong negative correlation between the total phytoplankton and large zooplankton [12].

CWP occurrence alters the nutrient concentrations and zooplankton community in an aquatic environment, and depending on the secchi depth, it can last for two or more weeks [13]. Moreover, the CWP is primarily observed with phytoplankton peaks occurring from late spring to summer, mainly because these resources are depleted after the occurrence of diatoms, which use the nutrients available from melting ice after winter [2].

Lake Paldang, an artificial lake created by the construction of a dam to utilize water resources, is greatly influenced by the seasonal climate, involving variable natural hydraulic and hydrological phenomena, such as typhoons, with localized heavy rains during the summer and drought in the spring and autumn. Environmental changes caused by these phenomena impact the aquatic environment and biomass, depending on the duration, size, and frequency [14,15]. Lake Paldang exhibits characteristics of a river-type lake because the reservoir area is small compared with the basin area, with its short hydraulic retention time (HRT) of 2.6–9.0 d that creates minor differences in the water quality by layer [16,17]. Thus, the lake is characterized as a complex dual water system, exhibiting the characteristics of a river depending on precipitation and the short HRT in the summer linked to high discharge, and as a lake during the dry season associated with low discharge [18]. Considering that Lake Paldang is an artificial confluence of the Namhan, Bukhan, and Gyeongan streams which involve different discharges, the boundaries between the inflow of the rivers and the lake remain unclear. In addition, because dams are present midstream and upstream of the Namhan and Bukhan Rivers, the water velocity and discharge can vary significantly, thereby affecting Lake Paldang [19]. The biological characteristics of this artificial lake are primarily influenced by hydrological phenomena [20,21], for example, the zooplankton community varies with the river inflow characteristics [21]. Therefore, Lake Paldang is expected to exhibit variations in CWP occurrence because it involves varied aquatic environments.

The aim of the present study was to highlight the occurrence patterns and causes of the CWP in Lake Paldang using hydrological, precipitation, physicochemical, and biological factors. Investigations were conducted at four sites representative of the dual water system (river and lake) to assess the association between the dual system and CWP characteristics.

## 2. Materials and Methods

### 2.1. Study Sites and Period

The study area, Lake Paldang, is an artificial confluence of the Namhan, Bukhan, and Gyeongan streams, serving as a multi-purpose dam for hydropower generation, a river maintenance water supply source, and for other functions including recreation. The lake basin covers an area of approximately 23,800 km^2^, with the Namhan, Bukhan, and Gyeongan stream areas accounting for 60%, 37%, and 3%, respectively. Considering its surface area of 36.5 km^2^ and the associated basin to reservoir area ratio of up to 652, the lake is vulnerable to pollutant inflow. The lake is shallow at the river inlets and deepest at the dam fronts, with an average depth of 6.4 m and maximum depth of 22 m, and an average hydraulic retention time (HRT) of approximately 7 d. The study sites were as follows: an area near the Paldang Dam (St. 1), exhibiting lake characteristics and the highest water depth; the Namhan River confluence area (St. 2), characterized by inflow with river characteristics; the Bukhan River confluence area (St. 3); and the Gyeongan stream confluence area (St. 4), characterized by a relatively small and shallow area. Sampling was performed once a week (excluding periods characterized by freezing) from March 2015 to July 2017 at average depths of 21, 9, 8, and 2.5 m for St. 1, 2, 3, and 4, respectively (Figure 1).

### 2.2. Precipitation and Hydrological Data

The precipitation data were obtained from daily observation records of the Yangpyeong weather station of the Korea Meteorological Administration, while hydrological data including the inflow, outflow, and water level were acquired from the Korea Water Management Information System (WAMIS: http://www.wamis.go.kr/, accessed on 2 April 2021). A total of 1,096 sets of daily hydrological data for 3 years were used. Factors with missing values due to snowy weather were excluded from the data before analysis. 

### 2.3. Sampling and Data Analysis

#### 2.3.1. Physicochemical Factors

The physicochemical factors measured during each sampling comprised the water temperature, pH, dissolved oxygen (DO), electrical conductivity (EC), turbidity, and secchi depth. The turbidity and secchi depth were determined in situ via a water quality instrument (YSI-EXO, YSI Inc., Yellow Springs, OH, USA) and the Secchi disk, respectively. To measure other parameters, each sample was prepared by mixing the surface layer (0–0.5 m from the surface layer), middle layer (middle of the maximum depth), and lower layer (0.5–1.0 m above the bottom of the maximum depth) from each site in equal proportions and transported to the laboratory. They were analyzed according to the standard method for Water Pollution Examination of the Ministry of Environment [22]. The biological oxygen demand (BOD), chemical oxygen demand (COD), suspended solids (SS), total nitrogen (TN), total phosphorus (TP), dissolved total nitrogen (DTN), ammonia nitrogen (NH_4_–N), nitric acid nitrogen (NO_3_–N), dissolved total phosphorus (DTP), orthophosphate as phosphorus (PO_4_–P), and Chl-*a* were determined in the laboratory. A total of 104 sets of weekly physicochemical data were used for each site.

#### 2.3.2. Biological Factors

Zooplankton were collected by filtering 10 L of the water through a 60-µm mesh, the collected sample was placed in a polystyrene container, and the concentration was raised to 5% using a neutral formalin solution (20%). The fixed sample was left to stand for at least 24 h to allow for precipitation, the supernatant was removed, the sample volume was adjusted to 10 mL, and then the sample was analyzed by microscopy. To analyze the phytoplankton, surface water samples were collected from the sites. The samples were preserved by adding a Lugol’s iodine solution (final concentration 0.3%). A preserved sample (1 mL) was allowed to settle in a Sedgewick–Rafter counting chamber (30 min) and then analyzed by microscopy. The quantitative and qualitative analyses of phyto-zooplankton samples were performed using a phase-contrast microscope (Nikon Eclipse; Nikon, Tokyo, Japan) at magnifications varying from 100× to 1000×. We used a Sedgewick–Rafter counting chamber, and the number of zooplankton individuals was converted to number of individuals per liter (Ind. L^−1^). Phytoplankton cell density was quantified as the number of cells per milliliter (cells. mL^−1^) of samples. Details of zooplankton identification are available in Segers [23], [24] for Rotifera; Chang and Min [25] for Copepoda; and Jeong, et al. [26] for Cladocera. Phytoplankton were identified at the genus or species level as described by John, et al. [27] and Wehr, et al. [28]. 

#### 2.3.3. CWP Occurrence Assessment

To assess the CWP occurrence at each site in Lake Paldang, variations in the secchi depth, turbidity, Chl-*a*, and zooplankton individuals were analyzed. According to the definition of CWP in Dröscher, et al. [1], a two-week period characterized by a continuous increase in secchi depth, a decrease in Chl-*a*, and an increase in individual zooplankton were considered which indicated the CWP occurrence conditions, while a decrease in turbidity was included for the final determination.

#### 2.3.4. Statistical Analysis

To analyze the differences before and after the CWP occurrence in Lake Paldang, a paired t-test was performed using data for each site, with *p* < 0.05 considered as statistically significant. Further, the relationships between the physicochemical and biological factors associated with the CWP occurrences were evaluated using Pearson’s correlations. The SPSS 20 program (IBM Corp., Armonk, NY, USA) was used for the statistical processing, and the significance was confirmed at *p* < 0.05. 

#### 2.3.5. Analysis of Environmental Factors Affecting CWP Occurrence Using the Network Model

In the present study, environmental factors affecting the occurrence of CWP in Lake Paldang were identified, and the sensitivity and influence of various factors evaluated using the Bayesian model. The analysis comprised data for precipitation, water temperature, turbidity, secchi depth, HRT, SS, water quality (TN, TP, PO_4_–P, NH_4_–N, DTN, and DTP), phytoplankton community (cyanobacteria, diatoms, green algae, and other algae) density, Chl-*a*, individual zooplankton community (Copepoda, Cladocera, and Rotifera), and RA (%) of zooplankton. All weekly sampling data from 2015 to 2017 were analyzed, excluding periods when sampling was impossible because of weather conditions. The statistical analysis results were normalized through the min–max normalization, followed by processing using the Bayesian search algorithm of the GeNIe modeler (GeNIe model, version 2.4.4601.0, BayesFusion, LLC, Pittsburgh, PA, USA). Sensitivity and influence evaluations were conducted using data with statistical significance (*p*-value) between 0.005 and 0.05.

For the Bayesian search algorithm analysis, the model operation background data were divided into Tiers 1–6. Tiers indicate the temporal order of the variables. In the Bayesian network model, there will be no retrogressive relationship between variables that occur later in time (in higher tiers) and nodes occurring earlier in time (in lower tiers) [29]. In this study, each tier was separated into an ecological relationship, which has been proved in ecological statistical studies such as correlation analysis [30,31,32]. These included the following: precipitation and water temperature in Tier 1; NH_4_–N, PO_4_–P, DTP, DTN, and HRT in Tier 2; phytoplankton (cyanobacteria, diatoms, green algae, and other algae) density and individual zooplankton (Cladocera, Copepoda, and Rotifera) in Tier 3; total phytoplankton cell density, total individual zooplankton, and RA of Copepoda, Cladocera, and Rotifera in Tier 4; TN, TP, and Chl-*a* in Tier 5; and SS concentration, secchi depth, and turbidity in Tier 6 (Table 1). The maximum of nodes connected from each factor was set to eight, while secchi depth and turbidity were set as targets of the algorithm. The sample size of the model was set to 50, and the k-fold cross-validation was applied to improve the model accuracy.

## 3. Results and Discussion

### 3.1. Precipitation and Hydrological Characteristics

In the period from January 2015 to December 2017, the total precipitation in the Lake Paldang basin was 2924.1 mm, including 800.3 mm in 2015, 935.9 mm in 2016, and 1187.9 mm in 2017, which indicated an increasing trend since 2014 (790.1 mm) as shown in Figure 2. The precipitation during summer (June to August) was 1,843.0 mm, which indicated a monsoon climate with 63.0% of the total precipitation concentrated in the summer. During the study period, the average water level of Lake Paldang was 25.2 m above sea level, with a maximum of 25.4 m and a minimum of 24.8 m, indicating minor water level fluctuations. The average inflow and outflow were 301.7 m^3^ s^−1^ and 301.6 m^3^ s^−1^, respectively, showing that a constant water level was maintained by controlling the sluice gate.

The average HRT of Lake Paldang during the study period was 14.5 d, involving 12.3 d in spring (March–May), 10.9 d in summer (June–August), 16.3 d in autumn (September–November), and 18.5 d in winter (December–-February), with the shortest HRT exhibited during summer. The average HRT was 16.6 d in 2015, 14.3 d in 2016, and 12.7 d in 2017, showing a decreasing trend. Therefore, because of the drought caused by low precipitation in 2014 (790.1 mm) and the outflow of the upstream dams, the inflow to Lake Paldang decreased and HRT increased (Figure 2). The HRT exhibited significant correlations (*p* < 0.05) with Copepoda individuals and Rotifera (negative), Cladocera (positive), and Copepoda (positive) RA as shown in Appendix A. Hydraulic changes in the water bodies are linked to heavy rainfall in Lake Paldang, which lies in a monsoon climate region [33]. Owing to the outflow of the upstream dams, the HRT decreased and zooplanktons were flushed, thereby impacting the individuals [34,35]. Further, secchi depth and turbidity showed significant correlations (*p* < 0.01) with precipitation and the HRT, indicating that HRT changes caused by precipitation could terminate a CWP (Appendix A). 

### 3.2. CWP Occurrence Pattern

CWP occurrence during the study period was characterized by low turbidity and Chl-*a*, though the period of increased secchi depth and zooplankton individuals, the occurrence frequency, and intensity varied between the study sites. The frequency of CWP occurrence was as follows: 12 occurrences of 2 weeks and 1 occurrence of 3 weeks in St. 1, which was the deepest among the sites, whereas 10 occurrences of 2 weeks and 1 of 3 weeks were recorded in St. 2, exhibiting strong lake-like characteristics. In total, 5 occurrences of 2 weeks and 1 of 3 weeks were observed in St. 3, with its strong river-like characteristics, whereas 12 occurrences of 2 weeks and 1 involving 3 weeks were observed in St. 4, characterized by its relatively shallow depth and long HRT. During the CWP events, the secchi depth increased by an average of 0.7, 0.9, 0.9, and 0.5 m in depth ranges of 1.2–5.0 m in St. 1, 1.2–5.9 m in St. 2, 1.5–4.6 m in St. 3, and 0.8–2.5 m in St. 4, respectively (Figure 3). The CWP primarily occurred from late spring to early summer and autumn, with few occurrences during the summer. This was possibly because as the temperature rises, Rotifera dominates Cladocera, thereby decreasing CWP events [36].

The turbidity decreased by an average of 0.9 (0.6–2.6 NTU) in St. 1, 1.0 (0.1–4.5 NTU) in St. 2, 1.3 (0.1–2.0 NTU) in St. 3, and 2.4 NTU (0.7–8.3 NTU) in St. 4. The highest average decrease in Chl-*a* was 15.5 mg m^−3^ in St. 2, followed by St. 4 (11.4 mg m^−3^), St. 1 (6.0 mg m^−3^), and St. 3 (3.6 mg m^−3^) (Figure 3). 

Regarding the changes in the individual zooplankton population, St. 4 showed the highest average increase (216.7 Ind L^−1^), followed by St. 2 (146.9 Ind L^−1^), St. 1 (100.6 Ind L^−1^), and St. 3 (5.2 Ind L^−1^). Cladocera (*Daphnia*), which are closely associated with the CWP through filter feeding, showed the highest increase in St. 2 (36.6 Ind L^−1^), followed by St. 4 (20.1 Ind L^−1^), St. 1 (13.2 Ind L^−1^), and St. 3 (5.2 Ind L^−1^). The individuals (*r* = 0.185) and RA (*r* = 0.250) of Cladocera exhibited significant positive correlations (*p* < 0.01) with the water temperature (Appendix A). This indicated that the water temperature rise in spring promoted the emergence and growth of Cladocera as well as influenced CWP occurrence by retarding the emergence period of predators [37]. 

### 3.3. Time Characteristics for CWP Occurrence

In the present study, Pearson’s correlation test was performed to assess the relationships between the secchi depth and environmental and biological factors associated with the CWP occurrence. The paired t-test was also conducted to evaluate the differences in the characteristics of each site before and after the CWP, as shown in Table 2. Secchi depth, turbidity, Chl-*a*, and zooplankton individuals, which are determinants of CWP occurrence in Lake Paldang, exhibited significant correlations among different factors, with turbidity and secchi depth showing a strong negative correlation (*r* = −0.497, *p* < 0.01). Further, the secchi depth, turbidity, and Chl-*a* showed significant differences (*p* < 0.05) after the CWP occurrence compared to before, for all sites, while the zooplankton individuals exhibited significant differences (*p* < 0.05) for all sites, except St. 3 (*p* = 0.122). At St. 3 (Bukhan River confluence area), the velocity and discharge were significantly affected by the control of the upstream dam sluice gate [19], which possibly impacted the zooplankton individuals, considering the significant difference (*p* < 0.05) in HRT before and after the CWP occurrences.

The total zooplankton individuals exhibited a negative correlation with secchi depth (*p* < 0.05); however, this result possibly reflected the correlation between Rotifera individuals and secchi depth (*p* < 0.01) because the increase in spring Rotifera showed a positive correlation with Chl-*a* (*p* < 0.01). This is consistent with the increase in Rotifera individuals and Chl-*a* in Lake Paldang reported by You, et al. [38] and Sim, et al. [39]. Among the zooplankton, Cladocera, a filter feeder, is commonly associated with the CWP [1], while the total zooplankton individuals exhibited a negative correlation with secchi depth. However, the Cladocera individuals and RA showed significant correlations (*p* < 0.01) with the secchi depth, thereby supporting the findings of previous studies indicating that Cladocera (*Daphnia*) is an important factor associated with the CWP (Appendix A) [40,41,42]. Secchi depth and turbidity were significantly correlated (*p* < 0.05) with nutrients; in particular, TP exhibited a strong negative correlation (*p* < 0.01) with secchi depth (*r* = −0.567) and strong positive correlation with turbidity (*r* = 0.764), indicating that the nutrient concentration decreased when the CWP occurred. In St. 1 and St. 4, which involved higher lake characteristics compared with St. 2 and St. 3, nutrients (TN and TP) differed significantly before and after the CWP occurrences, indicating high nutrient concentrations in St. 1 and St. 4 before CWP events. Regarding the strong correlation among the CWP, secchi depth, and nutrients, Winder and Schindler [43] noted that a high TP concentration hinders CWP occurrence and phytoplankton emergence in spring. Meanwhile, Thackeray, et al. [44] suggested that a high TP concentration can accelerate the emergence time of phytoplankton, thereby increasing *Daphnia* population which feed on phytoplankton.

Luecke, et al. [45] suggested that *Daphnia* biomass can rapidly increase under low temperature and high food availability conditions. Moreover, according to Matsuzaki, et al. [42], at low water temperature, TP influences the emergence time and biomass of phytoplankton, thereby indirectly affecting the CWP occurrence. Therefore, the occurrence of CWP in Lake Paldang triggered the emergence of edible phytoplankton (small edible cells) because of the increase in nutrients in spring [9,10], which promoted the emergence of the zooplankton (Cladocera). Consequently, Chl-*a* and turbidity decreased, while secchi depth increased, and eventually, CWP occurred. 

### 3.4. Analysis of CWP Occurrence Factors through the Network Model

Based on the zooplankton community in the Lake Paldang watershed, an ecological Bayesian network model between physio-chemical environmental factors and plankton communities was analyzed. In the Bayesian network, each site had a different connection from each other due to the hydrological characteristic of the Lake Paldang watershed. However, commonly in all the sites, the environment and plankton factors were closely connected with a variation of secchi depth. At the Lake Paldang watershed scale, which combined all site data, the variation of secchi depth was very sensitive to the densities of diatom and cyanobacteria. When all four sites of Lake Paldang were integrated and analyzed, the secchi depth was the most sensitive to the changes in diatoms and blue-green algae and turbidity, and turbidity was sensitive to precipitation, indicating that secchi depth is indirectly affected by precipitation (Figure 4). Moreover, although various chemical environmental factors (water quality) had relatively low sensitivity, secchi depth was connected to the N and P sources. However, individuals and RA of zooplankton were not related to secchi depth at the watershed scale. 

In the Bayesian network model of each site, the connection of factors reflected the hydraulics and environment characteristics of each site. St. 2 located in Namhan River which inflows to Lake Paldang exhibited a very similar network connection with St. 1 in Lake Paldang. However, St. 3 and St. 4 in Bukhan River and Gyeongan stream, respectively, which inflow to Lake Paldang, have independent network connections which revealed environmental characteristics that were different from St. 1. In St. 1 and 2, secchi depth was very sensitive to the RA of Cladocera, and these two factors had strong connection in the network (Figure 5 and Figure 6). Moreover, the RA of the Cladocera was also linked with Copepoda individuals, and the RA of Cladocera was affected by individuals of Copepoda and Cladocera.

Although the total zooplankton individuals were significantly affected by various Rotifera individuals, the Cladocera individuals and secchi depth were not directly connected in the network with Rotifera individuals. However, secchi depth in St. 2 was connected with the RA of Rotifera, and the Rotifera was affected by individuals of Cladocera and Copepoda. Therefore, the Rotifera affected the RA of Cladocera, and secchi depth was very sensitive to the RA of the Rotifera. There was no direct connection with secchi depth and less sensitivity than zooplankton factors (Rotifera, Copepoda), while precipitation and some chemical factors (DTP, PO_4_–P, NH_4_–N) had sensitivity to secchi depth in St.1 and 2. Consequently, although secchi depth in St. 1 and 2 were not connected directly, secchi depth was sensitive to variations of zooplankton, particularly, Cladocera individuals and RA. This connection and sensitivity between Cladocera and secchi depth have primarily been observed in lakes with restricted feeding [46,47]. Therefore, the high secchi depth observed at St. 1 and 2 implied that the feeding action of Cladocera on phytoplankton led to the CWP.

In St. 3 and 4, secchi depth showed no direct connection to biological factors but was directly linked with water quality and meteorological factors (Figure 7 and Figure 8). In St. 3, located in the Bukhan River, secchi depth was highly sensitive to residence time, precipitation, and turbidity (Figure 7). In particular, precipitation affected residence time and turbidity, and consequently, HRT and turbidity directly affected secchi depth. According to Lehman, et al. [48], the CWP occurs when the HRT reaches 20 d. In St. 3, the HRT effect was more evident than other sites due to the hydraulic characteristics. The zooplankton, which comprised variations of Cladocera and Copepoda individuals, had an indirect connection with secchi depth; however, these had relatively low sensitivity to secchi depth compared to hydraulic and meteorological factors. In contrast to St. 3, only Cladocera individuals were indirectly connected with secchi depth in St. 4. In St. 3, most impacts on secchi depth were attributed to precipitation and HRT, whereas, in St. 4, secchi depth was connected to various chemical factors.

Among the water quality factors, only the PO_4_–P concentration was correlated to secchi depth, which was consistent with the difference in nutrients before and after the CWP occurrences revealed by the paired t-test. Conversely, secchi depth in St. 4 was more sensitive to water quality and meteorological factors (Figure 8). The effects of light and temperature depend on the water depth, and these factors can accelerate CWP occurrence through the interaction between phytoplankton and zooplankton [49]. Therefore, St. 4 was considered to be significantly affected by meteorological factors. Among biological factors, changes in Cladocera and phytoplankton (cyanobacteria, diatoms, and other algae) densities were linked to secchi depth variations, though no direct connection was observed to some biological factors. Although Cladocera formed no direct node with secchi depth and turbidity, the Cladocera individuals influenced the cyanobacteria cell density, while the density of cyanobacteria affected turbidity. The appearance of cyanobacteria among phytoplankton can influence the succession of Cladocera communities [50]; therefore, the secchi depth changes linked to this network were sensitive to the variation in Cladocera individuals.

Phytoplankton showed no direct network with secchi depth for all sites, but indirect nodes associated with turbidity were observed. Phytoplankton did not show a direct correlation with secchi depth because the sensitivity of phytoplankton factor was relatively low compared with that of the main factor affecting secchi depth in each site, as determined by modeling at each site. However, in the case of Figure 4, which was modeled for all sites, the density of cyanobacteria and diatoms was found to be related to secchi depth. This can be attributed to the relatively high sensitivity of phytoplankton factor among factors that can affect secchi depth in all sites.

In addition, instead of biological factors, water quality, meteorological, and hydrological factors formed nodes with secchi depth, and precipitation was found to be a factor affecting secchi depth at all sites. In addition to precipitation, the HRT also showed sensitivity at St. 3, whereas biological and water quality factors exhibited no sensitivity (Figure 5). Considering the water quality factors, except for St. 3, the DTP concentration generally affected secchi depth, while PO_4_–P influenced the secchi depth at St. 1 and St. 2.

The individuals and RA of each zooplankton taxon were not sensitive to the phytoplankton cell density, with the zooplankton abundance affecting that of phytoplankton only at St. 4 (Figure 8). At St. 1 and St. 2, which were characterized by high sensitivity between secchi depth and Cladocera RA, the latter was also sensitive to changes in Copepoda and Cladocera individuals (Figure 5 and Figure 6). In addition, changes in Copepoda and Cladocera individuals for St. 2 affected the Rotifera RA (Figure 6). At St. 3, secchi depth was sensitive to the Cladocera and Copepoda individuals but not to Rotifera (Figure 7).

## 4. Conclusions

In the present study, the CWP occurrence at four sites in Lake Paldang was investigated to assess various patterns. The occurrence period covered the entire year, from March to June. During the study duration, the CWP occurred 13 times in St. 1 (dam front) and 11 times in St. 2, which exhibited strong lake characteristics, seven times in St. 3, which showed strong river characteristics, and 14 times in St. 4, characterized by a relatively shallow depth and long HRT. Thus, the CWP was frequent at sites with lake characteristics and long HRT. According to analysis of the relationships between environmental factors and secchi depth, turbidity, zooplankton individuals, and Chl-*a*, which were the main factors associated with CWP occurrence, the hydraulic and hydrological factors of precipitation and HRT exhibited significant correlations (*p* < 0.01) with secchi depth and turbidity. This indicated that a change in the HRT because of precipitation could influence CWP occurrence. Further, among the zooplankton associated with the CWP, the Cladocera individuals and RA showed significant correlations (*p* < 0.01) with secchi depth and the water temperature. This indicated that the emergence of Cladocera because of the rising water temperature in spring could cause the CWP. 

The sensitivity and influence of different factors at each site were investigated using Bayesian models, and the results reveal differences between the river-type site (St. 3) and lake-type sites (St. 1 and 2). The relationship between Cladocera RA and secchi depth exhibited a higher sensitivity for the lake-type than river-type sites. Although sensitivity to precipitation and HRT was high for the river-type site, the lake-type sites, which were advantageous for the emergence of Cladocera and characterized by long HRT, displayed higher favorability for CWP occurrence. The data for St. 4 showed that the occurrence of the CWP was associated with the interaction between zooplankton and phytoplankton and the accompanying decrease in turbidity.

Although CWP occurrence was prominent in the spring, the present study revealed that site-specific CWP could occur throughout the year, regardless of the season. Such an occurrence is commonly characterized by a stable water body based on the water system characteristics, accompanied by an increase in zooplankton (Cladocera), a decrease in the phytoplankton biomass, an increase in secchi depth, and a decrease in turbidity.

## Figures and Tables

**Figure 1 ijerph-18-07205-f001:**
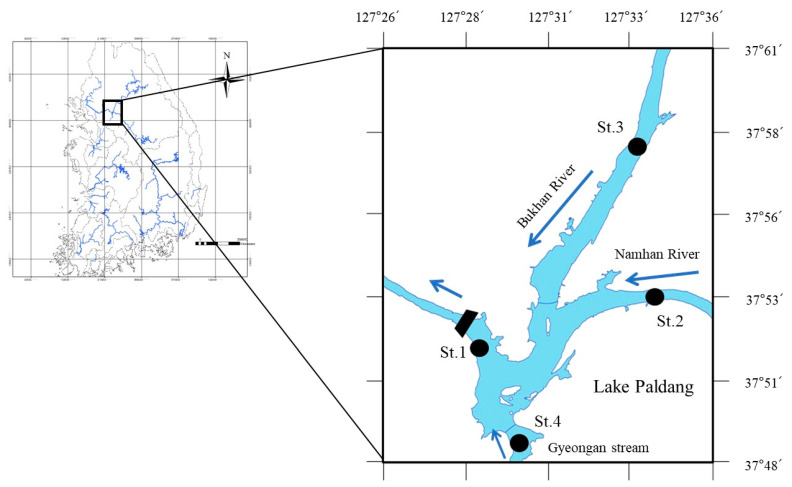
Map showing the location of Lake Paldang and the study sites including: St. 1, Lake Paldang; St. 2, Namhan River; St. 3, Bukhan River; and St. 4, Gyeongan stream.

**Figure 2 ijerph-18-07205-f002:**
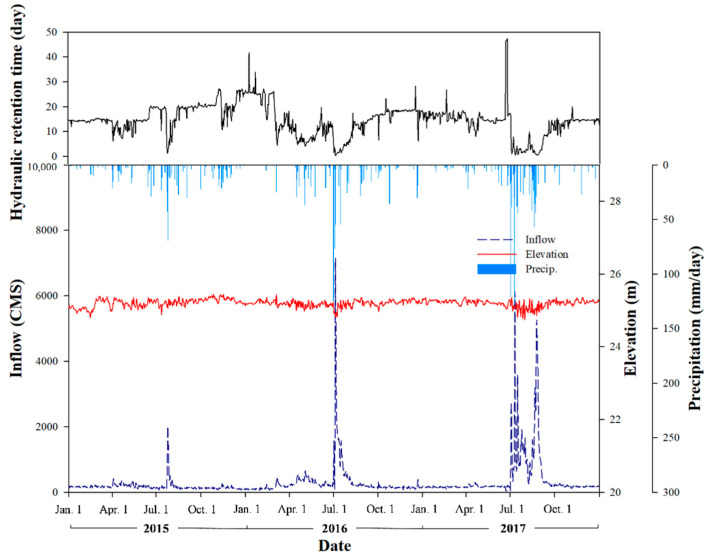
Seasonal changes in the inflow, elevation, precipitation, and hydraulic retention time (HRT) for Lake Paldang from 2015 to 2017.

**Figure 3 ijerph-18-07205-f003:**
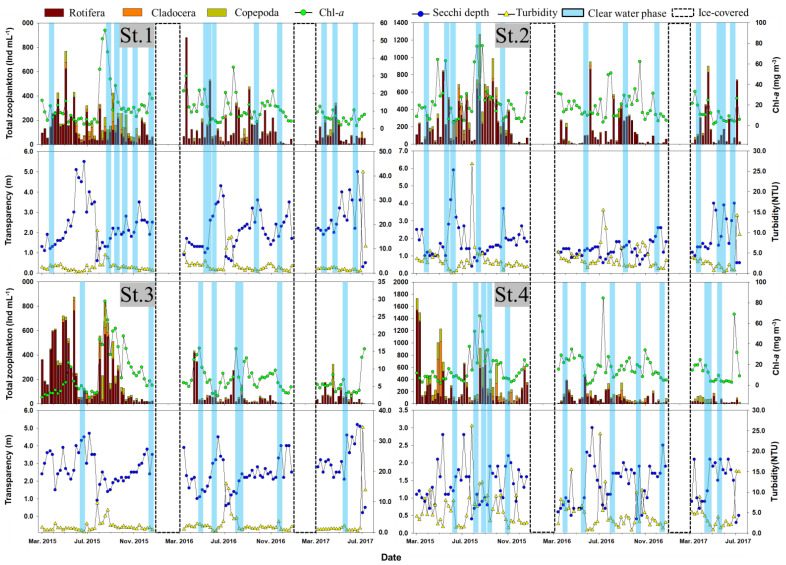
Environmental variables showing significant differences among the four sites in Lake Paldang studied from March 2015 to July 2017.

**Figure 4 ijerph-18-07205-f004:**
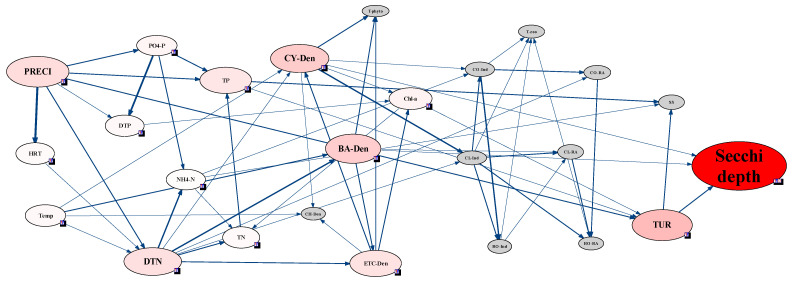
Sensitivity test and strength of influence analysis based on the Bayesian network approach for assessing the clear-water phase (CWP) in the Lake Paldang watershed. The Bayesian network data were discretized using a hierarchical method and bin count 3. The data were collected from four sites in the Lake Paldang watershed, Namhan River, and Bukhan River.

**Figure 5 ijerph-18-07205-f005:**
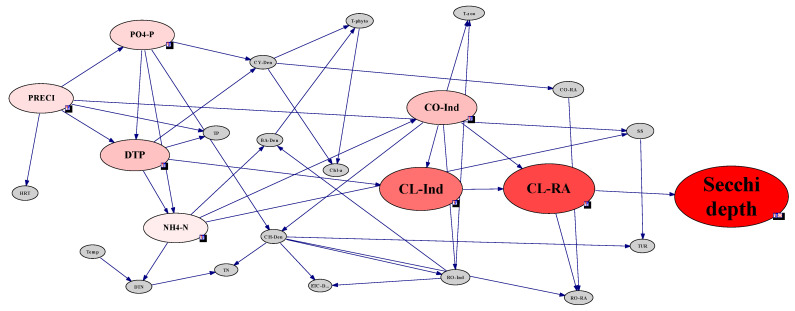
Sensitivity test and strength of influence analysis based on the Bayesian network approach for assessing the CWP in St. 1 near the Paldang dam. All data for St. 1 used in the Bayesian network were discretized by a hierarchical method and bin count 3.

**Figure 6 ijerph-18-07205-f006:**
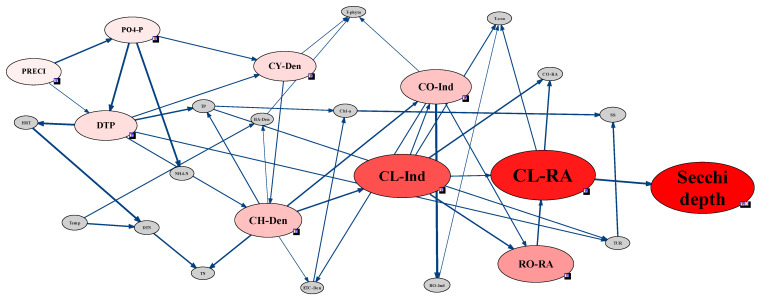
Sensitivity test and strength of influence analysis using the Bayesian network approach for assessing the CWP in St. 2, which was downstream of the Namhan River watershed. All data for St. 2 used in the Bayesian network were discretized using a hierarchical method and bin count 3.

**Figure 7 ijerph-18-07205-f007:**
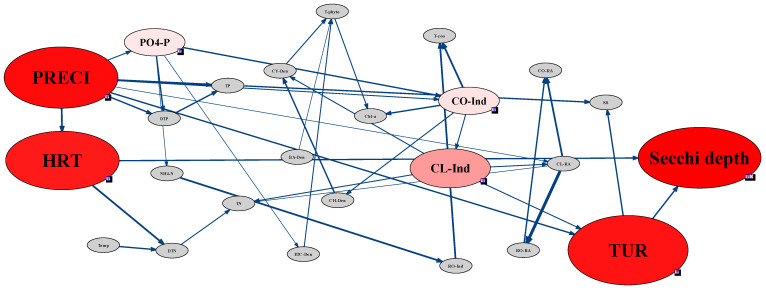
Sensitivity test and strength of influence analysis using the Bayesian network approach for assessing the CWP in St. 3, which was at the junction between the Bukhan River and Lake Paldang. All data for St. 3 used in the Bayesian network were discretized using a hierarchical method and bin count 3.

**Figure 8 ijerph-18-07205-f008:**
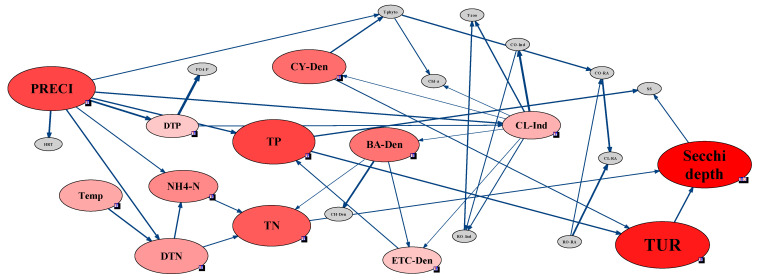
Sensitivity test and strength of influence analysis using the Bayesian network approach for assessing the CWP in St. 4, which was downstream of Gyeongan stream. All data for St. 4 used in the Bayesian network were discretized using the hierarchical method and bin count 3.

**Table 1 ijerph-18-07205-t001:** Temporal tier components of the Bayesian search algorithm.

Temporal Tier 1	Temporal Tier 2	Temporal Tier 3	Temporal Tier 4	Temporal Tier 5	Temporal Tier 6
PRECI	HRT	CY	T-phyto-Den.	TN	SS
Temp	NH_4_–N	BA	T-zoo-Ind.	TP	SD
	PO_4_–P	CH	RO-RA	Chl-*a*	TUR
	DTN	ETC-Den	CL-RA		
	DTP	RO-Ind	CO-RA		
		CL-Ind			
		CO-Ind			

PRECI: precipitation; Temp: water temperature; HRT: hydraulic retention time; NH_4_–N: ammonium nitrogen; PO_4_–P: orthophosphate as phosphorus; DTN: dissolved total nitrogen; DTP: dissolved total phosphorus; CY: cyanobacteria; BA: bacillariophyceae; CH: chlorophyceae; ETC: Euglenophyceae, Cryptophyceae, Chrysophyceae, Dinophyceae, Synurophyceae; RO: Rotifer; CL: Cladocera; CO: Copepod; T-phyto: total phytoplankton; T-zoo: total zooplankton; TN: total nitrogen; TP: total phosphorus; SS: suspended solids; SD: secchi depth; TUR: turbidity; Ind: individuals; RA: relative abundance; Den: cell density.

**Table 2 ijerph-18-07205-t002:** Data for paired t-test on variable factors before and after the clear-water phase (CWP) in Lake Paldang.

Parameter (Unit)	Site	Before the CWP	After the CWP	t	*p*
Secchi depth (m)	St. 1	1.9 ± 0.5	2.6 ± 0.9	−3.763	0.00271 *
	St. 2	1.7 ± 1.0	2.7 ± 1.5	−3.969	0.00265 *
	St. 3	2.4 ± 1.1	3.4 ± 1.1	−3.417	0.01419 *
	St. 4	1.0 ± 0.4	1.7 ± 0.5	−5.078	0.00021 *
Turbidity (NTU)	St. 1	2.7 ± 1.4	1.7 ± 0.6	3.204	0.00758 *
	St. 2	2.6 ± 1.2	1.6 ± 1.3	4.468	0.00120 *
	St. 3	2.4 ± 1.4	0.8 ± 0.6	3.234	0.01782 *
	St. 4	6.3 ± 2.9	3.6 ± 2.5	4.873	0.00030 *
Total zooplankton (Ind L^−1^)	St. 1	96.4 ± 67.5	196.6 ± 138.0	−5.838	0.00008 *
	St. 2	166.9 ± 210.3	321.4 ± 347.7	−2.410	0.03671 *
	St. 3	36.3 ± 22.7	72.0 ± 53.9	−1.797	0.12243
	St. 4	128.7 ± 152.2	288.5 ± 308.8	−2.846	0.01377 *
Chlorophyll-*a* (mg m^−3^)	St. 1	15.7 ± 10.0	9.2 ± 7.2	4.293	0.00105 *
	St. 2	27.3 ± 22.8	10.3 ± 8.1	3.049	0.01227 *
	St. 3	9.0 ± 4.7	4.9 ± 2.6	2.842	0.02948 *
	St. 4	23.2 ± 14.9	11.5 ± 7.1	4.409	0.00071 *
Total nitrogen (mg L^−1^)	St. 1	1.898 ± 0.470	1.799 ± 0.385	2.335	0.03774 *
	St. 2	1.997 ± 0.456	1.963 ± 0.448	0.444	0.66658
	St. 3	1.805 ± 0.285	1.818 ± 0.339	−0.256	0.80627
	St. 4	2.483 ± 1.073	2.133 ± 0.851	3.725	0.00255 *
Total phosphorus (mg L^−1^)	St. 1	0.021 ± 0.008	0.016 ± 0.004	2.759	0.01733 *
	St. 2	0.031 ± 0.018	0.026 ± 0.014	2.159	0.05617
	St. 3	0.014 ± 0.007	0.012 ± 0.003	1.451	0.19707
	St. 4	0.031 ± 0.010	0.023 ± 0.009	3.934	0.00171 *
Hydraulic retention time (d)	St. 1	15.8 ± 4.7	16.9 ± 5.3	−1.607	0.13396
	St. 2	13.9 ± 3.5	16.7 ± 6.3	−1.652	0.12964
	St. 3	12.2 ± 6.1	14.9 ± 6.5	−2.447	0.04997 *
	St. 4	13.7 ± 4.9	14.4 ± 4.8	−1.417	0.17995

* *p* < 0.05 (95% confidence level).

## Data Availability

Not applicable.

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
