# Peer review of "Influence of Zooplankton and Environmental Factors on Clear-Water Phase in Lake Paldang, South Korea"

_ijerph, 2021, doi:10.3390/ijerph18137205_

Round 1

Reviewer 1 Report

Dear Authors,

Below are my comments to the text:
I am generally against the use of abbreviations other than those that are well known and commonly used. New shortcuts should not be created - they are confusing. I mean e.g. temp = temperature (not WTE); Water transparency (SD) = Secchi depth (not TRAN)  etc… it is about information in the text and table.
The samples were taken from different depths - "depths of 21, 9, 8, and 2.5 m ..." - please specify whether these are samples taken from the surface to the bottom of the reservoir - it is not clear - whether these are the maximum depths of these sites. The examined stations were varying greatly in the case of maximal depth. This may influence the obtained results and their interpretation. the average of Phytoplankton abundance should be lower in deep lakes compared to shallow ones ( light can penetrate to the bottom). There is a question-Whether the concentration of chlorophyll was averaged from the entire vertical section or from the euphotic zone - it is worth clarifying.
 “Phytoplankton showed no direct network with transparency” (line 369)- this conclusion seems quite surprising –it would be great to get some more comments about this
Line 148 - "phytoplankton community (cyanobacteria, diatoms, green algae, and other algae) density,"- the methodology did not specify how the taxonomic composition of phytoplankton was analyzed. It is not clear whether the phytoplankton components were analyzed and with what methods - the results are presented - please specify it in the methodology, e.g. "phytoplankton cell density" - it was not specified with what methods it was analyzed - whether the same chamber was used as in the case of zooplankton analyzes.

I also noticed a few minor editorial errors in the text, such as:
Fig. 3 “transparency” and “turbidity”lines were explained twice (once in the chart and once in the legend) - this elements should be removed from the chart (St. 2)
Taxa names should be in italics, eg Daphnia -line 276

Author Response

Response to Reviewer 1

Specific comments

Below are my comments to the text:

Q1. I am generally against the use of abbreviations other than those that are well known and commonly used. New shortcuts should not be created - they are confusing. I mean e.g. temp = temperature (not WTE); Water transparency (SD) = Secchi depth (not TRAN)  etc… it is about information in the text and table.

Answer: Thank you for the helpful comment. As the reviewer suggested, we have changed the abbreviations (Table 1) in the revised manuscript as follows;

- From “WTE” to “Temp”

- From “TRAN” to “SD”,

In addition, all “transparency” changed to “secchi depth” in the revised manuscript including the Figure 4 - 8.  

Q2. The samples were taken from different depths - "depths of 21, 9, 8, and 2.5 m ..." - please specify whether these are samples taken from the surface to the bottom of the reservoir - it is not clear - whether these are the maximum depths of these sites. The examined stations were varying greatly in the case of maximal depth. This may influence the obtained results and their interpretation. the average of Phytoplankton abundance should be lower in deep lakes compared to shallow ones ( light can penetrate to the bottom). There is a question-Whether the concentration of chlorophyll was averaged from the entire vertical section or from the euphotic zone - it is worth clarifying.

Answer: Thank you for the comment. We have added the “how the water sample was made” in the revised manuscript as follows;   

“each sample was prepared by mixing the surface layer (0-0.5m from the surface layer), middle layer (middle of the maximum depth), and the lower layer (0.5-1.0 m above the bottom of the maximum depth) from each site in equal proportions.” (Line 112-115)

Q3. “Phytoplankton showed no direct network with transparency” (line 369)- this conclusion seems quite surprising –it would be great to get some more comments about this

Line 148 - "phytoplankton community (cyanobacteria, diatoms, green algae, and other algae) density,"- the methodology did not specify how the taxonomic composition of phytoplankton was analyzed. It is not clear whether the phytoplankton components were analyzed and with what methods - the results are presented - please specify it in the methodology, e.g. "phytoplankton cell density" - it was not specified with what methods it was analyzed - whether the same chamber was used as in the case of zooplankton analyzes.

Answer: Thank you for the comment. The reasons why phytoplankton did not show a direct correlation with secchi depth are as follows;

“Phytoplankton did not show a direct correlation with secchi depth because the sensitivity of phytoplankton factor was relatively low compared with that of the main factor affecting secchi depth in each site, as determined by modeling at each site. However, in the case of Fig. 4, which was modeled for all sites, the density of cyanobacteria and diatoms was found to be related to secchi depth. This can be attributed to the relatively high sensitivity of phytoplankton factor among factors that can affect secchi depth in all sites” (Line 392-398).

In addition, the analysis method of phytoplankton is shown in the revised manuscript (Line 127-139).

Q4. I also noticed a few minor editorial errors in the text, such as:

Fig. 3 “transparency” and “turbidity”lines were explained twice (once in the chart and once in the legend) - this elements should be removed from the chart (St. 2)

Taxa names should be in italics, eg Daphnia -line 276

Answer: Thank you for having us clarify this. As the reviewer suggested, we removed the legend in Fig. 3 (St.2). In addition, “Daphnia” has been changed to “Daphnia” in the revised manuscript.

We have addressed all the reviewer’s comments and corrections in the text and would like to thank them for his/her assistance.

Reviewer 2 Report

The manuscript describes the results of research on the conditions under which the clean water phase appears in an artificial dam reservoir. Research conducted for two years, at weekly intervals, in which many factors describing the physico-chemical properties of water as well as phyto- and zooplankton were taken into account, are interesting and the obtained results are valuable. In the assessment of the significance (significance) of individual variables, Bayesian networks were used, which allowed to illustrate the interrelationships between the studied variables. I believe that the work is valuable and worth publishing. However, I suggest that the manuscript should be revised to increase the precision and clarity of the narrative.

Line 22: "shallow depth / long HRT sites" I guess it is opposed to shallow and deep sites, but were the long sites to be the opposite of depth sites?

Lines 33-35: I cannot access Dröscher, et al. 2008, but already in the title, those authors state that "Daphnia control of the spring clear-water phase" in lakes with in polymictic lakes, which are characterized by different productivity and size. Are the conditions for the occurrence of this phase really defined there, and do they actually say that the condition for the occurrence of the pure water phase is its eminent feature - clean water (usually expressed as transparent water measured with a Secchi disc)? Already according to the second cited work, the conditions / circumstances of CWP occurrence are differently defined than in the first sentence of the manuscript.

Lines 39-40: I don't know of any particular reasons why shallow lake water should have a particularly larger light transmission. The colour of the water and the presence of any kind of suspension determine this quality of the water. The colour of the water in lakes is most often caused by humic substances, and the suspension is usually composed of either phytoplankton and dead phytoplankton cells or organic sediments torn from the bottom (due to waves). This last-mentioned process is most often observed in shallow (and extensive) lakes.

Lines 40-42: I would not use the phrase "edible cells (of phytoplankton) favours CWP occurrence" as the multiplication of fine phytoplankton usually leads to the opposite, i.e. water with low transparency. Described in the manuscript, CWP is the result of the effective control of phytoplankton by zooplankton, i.e. the presence (in sufficient numbers) of such zooplankton organisms that can effectively reduce the occurrence of usually small phytoplankton organisms. Fish that feed on zooplankton also play an important role here.

Lines 50-51: "[...] the CWP is primarily observed during late spring in eutrophic lakes, characterized by variable nutrient load forms and sizes" - In limnological literature, in the second half of the last century, attention was drawn to the characteristic phases of the phytoplankton seasonal development cycle. After the ice melts, using the available nutrients released in winter, when the water temperature is still relatively low and full spring circulation takes place, diatoms develop, and as these resources are depleted (in lakes with still moderate fertility) there is a clean water (CWP) phase followed by the development of the summer peak of phytoplankton. The more the reservoir is rich in nutrients (fertile), the smaller the amplitude of phytoplankton biomass changes, and the longer the periods with high biomass.

Line 52: This short description of Lake Paldang must be preceded in the main text by the clearly stated information that it is an artificial dam reservoir. Meanwhile, the reader learns about it only in the second half of this paragraph (lines 62-63). It must be corrected.

Line 72: Why is it emphasized here that hydraulic, hydrological […] factors were analyzed when in other places only hydrological and precipitation data (lines 98-100) are clearly visible.

Line 87: "The study sites included 86 the following: the front of the Paldang Dam (St. 1), ...",

instead, it is better to describe this position as in the caption for Figure 5 of  “St. 1 is located near the Paldang dam".

Lines 92-93: What does the statement "at average depths" of sampling in individual positions mean - - eg on ST. 1 from a depth of 21 m, or from a layer of 0-21 m. If, I suppose, the latter, it should be precisely described how such samples were collected. However, if samples collected from water of different thickness were compared, it should be shown that those collected from deeper places were homogeneous - from one constantly mixed layer of water.

Line 102: "Factors with… outliers were excluded from the data before analysis". What does it mean? After all, for example, sudden precipitation will be "outliers". Was such data also excluded? If a value was missing, then all other measured values were also ignored - factors?

Lines 103-127: The sections should clearly state how many sets of hydrological and physico-chemical data sets have been collected.

Lines 109-115: It must be clearly stated that the parameters listed in lines 111-115 were determined according to the research methods contained in the study mentioned in the text (with the title). I hope it is widely available (available on-line). There is no point in repeating the information that the analyzes were carried out in the laboratory (Line 110 and 115).

Important note: Why as many as two interchangeable parameters were included in the analyzes. Water turbidity are the optical properties of fine suspensions in a water sample that cause light to scatter. It is conditioned by the presence of various undissolved organic and inorganic compounds, e.g. it can be caused by the presence of clay, precipitating compounds of iron, manganese, aluminum, humic acids, plankton. Transparency, on the other hand, is like the inverse of turbidity, which is determined with the help of a white disc by measuring the depth of water at which it becomes invisible. Thus the defined transparency gives approximate results. Therefore, it is obvious that the values of these variables are closely correlated with each other, which must be reflected in networks, therefore emphasizing such a fact is of no importance for the conducted analyzes.

Lines 116-127: There is no description of how phytoplankton analyzes were performed. On the other hand, in the existing and missing description, it should also be specified what type of values were used in further analyzes, eg RA.

Lines 150-152: So how many "sets" of data (of how many weeks) were used in this analysis?

Lines 158-162: Clarifications to the table need to be improved:

- What does the asterisk mean?

- What is hidden under the names of the taxonomic groups Cyanobacteria, Bacillariophyceae, Chlorophyceae, The last two Latin names should be capitalized.

- ETC: - ETC-Den is in the table.

- RO; CL; WHAT. - RO-Ind, Cl-Ind, CO-Ind is in the table.

- ETC (Euglenophyceae, Cryptophyceae, Chrysophyceae, Dinophyceae, Synurophyceae) - Why were these separate systematic groups integrated in this way?

Lines 164-174: I don't understand the principle of dividing data into levels for analysis with the Bayerian model. On lines 146-149 another grouping of these data is indicated - more logical in my opinion. For the adopted method, therefore, it would be necessary to provide either the literature that shows that it should have been done this way, or some justification should be provided.

Line 179: Why from 2014 when the 2015-2017 calendar years were analyzed?

Line 179: Why from 2014 when the 2015-2017 calendar years were analyzed?

Lines 195-202; This piece of text should be inserted after any zooplankton results are given, plus more detailed information about correlations, e.g. positive-negative, strong-weak, some values, equations? (Regardless of the table in the supplementary material, which is a separate material from the text of the work).

Lines 204-205: The charts only show the variation in the values of the various variables. In order to be able to claim the significance of differences, such a chart should include information on the results of statistical tests.

Line 210-211; 212 and 215: "one occurrence of three weeks or more" - one period may have only a specific number of days of duration.

Lines 222-225: Description very difficult to read (incomprehensible).

Lines 273-275: What do the asterisks in the table mean?

Captions in figures 4-8 require supplementing the information about the meaning of the colors used (the degree of staining) and the size of the ellipses.

Lines 292-293: I do not see such a direct connection in this drawing. In the figure, transparency is directly related to only three variables: cyanobacterial abundance, Cladocerian abundance and turbidity. Indirectly through other variables with many others, but why only some of them are mentioned, I do not understand.

Author Response

Response to Reviewer 2

Specific comments

The manuscript describes the results of research on the conditions under which the clean water phase appears in an artificial dam reservoir. Research conducted for two years, at weekly intervals, in which many factors describing the physico-chemical properties of water as well as phyto- and zooplankton were taken into account, are interesting and the obtained results are valuable. In the assessment of the significance (significance) of individual variables, Bayesian networks were used, which allowed to illustrate the interrelationships between the studied variables. I believe that the work is valuable and worth publishing. However, I suggest that the manuscript should be revised to increase the precision and clarity of the narrative.

Q1. Line 22: "shallow depth / long HRT sites" I guess it is opposed to shallow and deep sites, but were the long sites to be the opposite of depth sites?

Answer: Thank you for the comment. As the reviewer suggested, we have changed to “shallow and deep” in the revised manuscript (Line 22).

Q2. Lines 33-35: I cannot access Dröscher, et al. 2008, but already in the title, those authors state that "Daphnia control of the spring clear-water phase" in lakes with in polymictic lakes, which are characterized by different productivity and size. Are the conditions for the occurrence of this phase really defined there, and do they actually say that the condition for the occurrence of the pure water phase is its eminent feature - clean water (usually expressed as transparent water measured with a Secchi disc)? Already according to the second cited work, the conditions / circumstances of CWP occurrence are differently defined than in the first sentence of the manuscript..

Answer: Thank you for the comment. We have changed the sentence in the revised manuscript as follows;

 “This is also consistent with the results of Sommer et al. [2].” (Line 35)

Q3. Lines 39-40: I don't know of any particular reasons why shallow lake water should have a particularly larger light transmission. The colour of the water and the presence of any kind of suspension determine this quality of the water. The colour of the water in lakes is most often caused by humic substances, and the suspension is usually composed of either phytoplankton and dead phytoplankton cells or organic sediments torn from the bottom (due to waves). This last-mentioned process is most often observed in shallow (and extensive) lakes.

Answer: Thank you for your kind suggestion. We have changed the sentence in the revised manuscript as follows;

“In particular, shallow and extensive lakes are characterized by high light transmission, except those heavily contaminated with natural or artificial organic matter, which increases the phytoplankton density” (Line 37-40).

Q4. Lines 40-42: I would not use the phrase "edible cells (of phytoplankton) favours CWP occurrence" as the multiplication of fine phytoplankton usually leads to the opposite, i.e. water with low transparency. Described in the manuscript, CWP is the result of the effective control of phytoplankton by zooplankton, i.e. the presence (in sufficient numbers) of such zooplankton organisms that can effectively reduce the occurrence of usually small phytoplankton organisms. Fish that feed on zooplankton also play an important role here.

Answer: Thank you for the comment. Edible cells, ie, fine phytoplankton, mean that filter feeding Cladocera increase. In the case of fish that feed on zooplankton, references were confirmed that it may affect the CWP, but, unfortunately, it was excluded because it was not investigated in this study.

Therefore, we have changed the sentence in the revised manuscript as follows;

“The occurrence of the CWP is favored by an increase in fine phytoplankton and an increase in biomass of the Cladocera order, which are mostly filter feeder” (Line 40-42).

Q5. Lines 50-51: "[...] the CWP is primarily observed during late spring in eutrophic lakes, characterized by variable nutrient load forms and sizes" - In limnological literature, in the second half of the last century, attention was drawn to the characteristic phases of the phytoplankton seasonal development cycle. After the ice melts, using the available nutrients released in winter, when the water temperature is still relatively low and full spring circulation takes place, diatoms develop, and as these resources are depleted (in lakes with still moderate fertility) there is a clean water (CWP) phase followed by the development of the summer peak of phytoplankton. The more the reservoir is rich in nutrients (fertile), the smaller the amplitude of phytoplankton biomass changes, and the longer the periods with high biomass.

Answer: Thank you for helpful comment. As the reviewer suggested, we have revised the sentence as follows;

“Moreover, the CWP is primarily observed with phytoplankton peaks occurring from late spring to summer, mainly because these resources are depleted after the occurrence of diatoms, which use the nutrients available from the melting ice after winter” (Line 49-52).

Q6. Line 52: This short description of Lake Paldang must be preceded in the main text by the clearly stated information that it is an artificial dam reservoir. Meanwhile, the reader learns about it only in the second half of this paragraph (lines 62-63). It must be corrected.

Answer: Thank you for the comment. We have clearly stated information about Lake Paldang as follows;

“Lake Paldang, an artificial lake created by the construction of a dam to utilize water resources” (Line 53-54).

Q7. Line 72: Why is it emphasized here that hydraulic, hydrological […] factors were analyzed when in other places only hydrological and precipitation data (lines 98-100) are clearly visible.

Answer: Thank you for the comment. We have changed the sentence as suggested by reviewer as follows;

“Lake Paldang using hydrological, precipitation, physicochemical, and biological factors” (Line 74-75)

We described physicochemical and biological factors in section of 2.3.1 and 2.3.2.

Q8. Line 87: "The study sites included 86 the following: the front of the Paldang Dam (St. 1), ...",

instead, it is better to describe this position as in the caption for Figure 5 of  “St. 1 is located near the Paldang dam".

Answer: Thank you for the comment. As the reviewer suggested, we have changed to “near the Paldang Dam (St.1)” (Line 89)

Q9. Lines 92-93: What does the statement "at average depths" of sampling in individual positions mean - - eg on ST. 1 from a depth of 21 m, or from a layer of 0-21 m. If, I suppose, the latter, it should be precisely described how such samples were collected. However, if samples collected from water of different thickness were compared, it should be shown that those collected from deeper places were homogeneous - from one constantly mixed layer of water.

Answer: Thank you for the comment. we have added the “how the water sample was made” in the revised manuscript as follows;  

“each sample was prepared by mixing the surface layer (0-0.5m from the surface layer), middle layer (middle of the maximum depth), and lower layer (0.5-1.0 m above the bottom of the maximum depth) from each site in equal proportions.” (Line 112-115)

Q10. Line 102: "Factors with… outliers were excluded from the data before analysis". What does it mean? After all, for example, sudden precipitation will be "outliers". Was such data also excluded? If a value was missing, then all other measured values were also ignored - factors?

Answer: Thank you for the comment. Lake Paldang has an icy season in winter. Therefore, it is impossible to collect water samples by boat at that time. The “Outlier” is a typo. Therefore, it has been modified as follows;

“Factors with missing values due to snowy weather were excluded from the data before analysis” (Line 104-105) .

Q11. Lines 103-127: The sections should clearly state how many sets of hydrological and physico-chemical data sets have been collected.

Answer: Thank you for the comment. The sets of hydrological and physico-chemical data are as follows;

“A total of 1,096 sets of daily hydrological data for 3 years were used” (Line 103-105).

“A total of 104 sets of weekly physicochemical data were used for each site” (Line 121).

Q12. Lines 109-115: It must be clearly stated that the parameters listed in lines 111-115 were determined according to the research methods contained in the study mentioned in the text (with the title). I hope it is widely available (available on-line). There is no point in repeating the information that the analyzes were carried out in the laboratory (Line 110 and 115).

Important note: Why as many as two interchangeable parameters were included in the analyzes. Water turbidity are the optical properties of fine suspensions in a water sample that cause light to scatter. It is conditioned by the presence of various undissolved organic and inorganic compounds, e.g. it can be caused by the presence of clay, precipitating compounds of iron, manganese, aluminum, humic acids, plankton. Transparency, on the other hand, is like the inverse of turbidity, which is determined with the help of a white disc by measuring the depth of water at which it becomes invisible. Thus the defined transparency gives approximate results. Therefore, it is obvious that the values of these variables are closely correlated with each other, which must be reflected in networks, therefore emphasizing such a fact is of no importance for the conducted analyzes.

Answer: Thank you for the comment.

Physicochemical factors are analyzed directly in our institution and the results are uploaded online. On the other hand, the weather data is the result of analysis by other agencies, so we refer to the online data.

I also agree with the reviewer that transparency and turbidity are interchangeable. However, our institute (Ministry of Environment) considers transparency and turbidity as important factors to reflect policy decisions. So we continue to produce and use both data. They are interchangeable, but not identical.

Q13. Lines 116-127: There is no description of how phytoplankton analyzes were performed. On the other hand, in the existing and missing description, it should also be specified what type of values were used in further analyzes, eg RA.

Answer: Thank you for the comment. As the reviewer suggested, we have described the analysis method of phytoplankton in the revised manuscript as follows;

“To analyze the phytoplankton, surface water samples were collected sites. The samples were preserved by adding a Lugol’s iodine solution (final concentration 0.3%). A preserved sample (1 mL) was allowed to settle in a Sedgwick–Rafter counting chamber (30 minutes), and then analyzed by microscopy. The quantitative and qualitative analyses of phyto-zooplankton samples were performed using a phase contrast microscope (Nikon Eclipse, Nikon, Tokyo, Japan) at magnifications varying from 100× to 1,000×. We used a Sedgwick-Rafter counting chamber, and the number of zooplankton individuals was converted to number of individuals per liter (Ind. L–1). Phytoplankton cell density was quantified as the number of cells per milliliter (cells. mL-1) of samples. Details on zooplankton identification are available in Segers [24], [25] for Rotifera; Chang and Min [26] for Copepoda, and Jeong, et al. [27] for Cladocera. Phytoplankton were identified at the genus or species level as described by John, et al. [28], Wehr, et al. [29]” (Line 127-139)

Q14. Lines 150-152: So how many "sets" of data (of how many weeks) were used in this analysis?

Answer: Thank you for the comment. We have added the sentence as suggested by the reviewer.

“A total of 104 sets of weekly physicochemical data were used for each site” (Line 121).

Q15. Lines 158-162: Clarifications to the table need to be improved:

- What does the asterisk mean?

- What is hidden under the names of the taxonomic groups Cyanobacteria, Bacillariophyceae, Chlorophyceae, The last two Latin names should be capitalized.

- ETC: - ETC-Den is in the table.

- RO; CL; WHAT. - RO-Ind, Cl-Ind, CO-Ind is in the table.

- ETC (Euglenophyceae, Cryptophyceae, Chrysophyceae, Dinophyceae, Synurophyceae) - Why were these separate systematic groups integrated in this way?

Answer: Thank you for the comment.

- The asterisk indicates abbreviation and have been corrected in the manuscript.

- Bacillariophyceae and Chlorophyceae have been revised in the manuscript.

- Below Table 1 is a description of each abbreviation (ETC and Den (cell density))

- Below Table 1 is a description of each abbreviation (RO: Rotifer, CL: Cladocera, RO-Ind: Rofiter individuals, CL-Ind: Cladocera individuals, CO-Ind: Copepod individuals)

- The sum of the Euglenophyceae, Cryptophyceae, Chrysophyceae, Dinophyceae and Synurophyceae order belonging to ETC was 14.6% of the total phytoplankton density, and it did not show any significant results or effects in statistical and model analysis, so it was combined into the ETC group and analyzed.

Q16. Lines 164-174: I don't understand the principle of dividing data into levels for analysis with the Bayerian model. On lines 146-149 another grouping of these data is indicated - more logical in my opinion. For the adopted method, therefore, it would be necessary to provide either the literature that shows that it should have been done this way, or some justification should be provided.

Answer: Thank you for the comment. As the reviewer suggested, we have added the sentence in the revised manuscript as follows;

“Tiers indicate the temporal order of the variables. In the Bayesian network model, there will be no retrogressive relationship between variables that occur later in time (in higher tiers) and nodes occurring earlier in time (in lower tiers) [30]. In this study, each tier was separated into an ecological relationship, which has been proved in ecological statistical studies such as correlation analysis [31-33]” (Line 178-182).

Q17. Line 179: Why from 2014 when the 2015-2017 calendar years were analyzed?

Answer: Thank you for the comment. We have added the precipitation in 2014 (790.1 mm) in the revised manuscript (Line 196 and 211)

Q18. Lines 195-202; This piece of text should be inserted after any zooplankton results are given, plus more detailed information about correlations, e.g. positive-negative, strong-weak, some values, equations? (Regardless of the table in the supplementary material, which is a separate material from the text of the work).

Answer: Thank you for the comment. As the reviewer suggested, we have modified the sentence in the revised manuscript as follows;

“The HRT exhibited significant correlations (p < 0.05) with Copepoda individuals and Rotifera (negative), Cladocera (positive), and Copepoda (positive) RA as shown in Table S1” (Line 212-214).

Q19. Lines 204-205: The charts only show the variation in the values of the various variables. In order to be able to claim the significance of differences, such a chart should include information on the results of statistical tests.

Answer: Thank you for the comment. The factors shown in Fig. 3 are factors highly correlated with CWP based on the Table S1 which is summarized to show the correlation between each factor.

Q20. Line 210-211; 212 and 215: "one occurrence of three weeks or more" - one period may have only a specific number of days of duration.

Answer: Thank you for the comment. As the reviewer suggested, we have revised the sentences as follows in the revised manuscript.

“12 occurrences of 2 weeks and 1 occurrence of 3 weeks in St. 1, which was the deepest among the sites, whereas 10 occurrences of 2 weeks and 1 of 3 were recorded in St. 2, exhibiting strong lake-like characteristics. 5 occurrences of 2 weeks and 1 of 3 weeks were linked to St. 3, with its strong river characteristics, while 12 occurrences of 2 weeks and 1 involving 3 weeks were observed at St. 4, characterized by its relatively shallow depth and long HRT” (Line 229-234).

Q21. Lines 222-225: Description very difficult to read (incomprehensible).

Answer: Thank you for the comment. As the reviewer suggested, we have changed the sentence in the revised manuscript as follows;

“The turbidity decreased by an average of 0.9 (0.6 – 2.6 NTU) in St. 1, 1.0 (0.1 – 4.5 NTU) in St. 2, 1.3 (0.1 – 2.0 NTU) in St. 3 and 2.4 NTU (0.7 – 8.3 NTU) in St. 4. The highest average decrease in Chl-a was 15.5 mg m–3 in St. 2, followed by St. 4 (11.4 mg m–3), St. 1 (6.0 mg m–3), and St. 3 (3.6 mg m–3)” (Line 240-243)  

Q22. Lines 273-275: What do the asterisks in the table mean?

Answer: Thank you for the comment. An asterisk means a 95% confidence level or higher (Line 293).  

Q23. Captions in figures 4-8 require supplementing the information about the meaning of the colors used (the degree of staining) and the size of the ellipses.

Answer: Thank you for the comment. The size and color of the circles indicate their relative sensitivity.

Q24. Lines 292-293: I do not see such a direct co nnection in this drawing. In the figure, transparency is directly related to only three variables: cyanobacterial abundance, Cladocerian abundance and turbidity. Indirectly through other variables with many others, but why only some of them are mentioned, I do not understand.

Answer: Thank you for the comment. We have changed the sentence in the revised manuscript as follows;

“When all four sites of Lake Paldang were integrated and analyzed, the secchi depth was most sensitive to the changes in diatoms and cyanobacteria and turbidity, and turbidity was sensitive to precipitation, indicating that secchi depth is indirectly affected by precipitation” (Line 311-314).

We have addressed all the reviewer’s comments and corrections in the text and would like to thank them for his/her assistance.

Reviewer 3 Report

The manuscript is in general clear and well written and increases the knowledge about a complex ecosystem both lacustrine and fluvial. In my opinion only minor changes are required before the publication.

- 2.3.2. Biological factors

You have divided zooplankton in its main groups Cladocera, Rotifera and Copepoda. Do you have species composition? Or at least genera composition? All Cladocera specimens were Daphnia? Not all Cladocera and Rotifera species are filter feeders, many of them are predators, such as the cladocerans Bythotrephes and Leptodora and the rotifers Asplanchna and Synchaeta. For rotifers it could be interesting verify if during the CWP there was a dominance in terms of biomass of microphagous species versus raptorial ones. (Obertegger, U., Smith, H. A., Flaim, G., & Wallace, R. L. (2011). Using the guild ratio to characterize pelagic rotifer communities. Hydrobiologia, 662(1), 157-162.    Obertegger, U., & Manca, M. (2011). Response of rotifer functional groups to changing trophic state and crustacean community.)

Lines 40-43: “The emergence of ….. order, which is a filter feeder [8-11]” Not all cladocera are filter feeders!

Lines 45, 229, 260, 271, 276: Daphnia in italic

Line 58: HRT, explain the meaning of the abbreviation

Author Response

Response to Reviewer 3

Specific comments

The manuscript is in general clear and well written and increases the knowledge about a complex ecosystem both lacustrine and fluvial. In my opinion only minor changes are required before the publication.

Q1. - 2.3.2. Biological factors

You have divided zooplankton in its main groups Cladocera, Rotifera and Copepoda. Do you have species composition? Or at least genera composition? All Cladocera specimens were Daphnia? Not all Cladocera and Rotifera species are filter feeders, many of them are predators, such as the cladocerans Bythotrephes and Leptodora and the rotifers Asplanchna and Synchaeta. For rotifers it could be interesting verify if during the CWP there was a dominance in terms of biomass of microphagous species versus raptorial ones. (Obertegger, U., Smith, H. A., Flaim, G., & Wallace, R. L. (2011). Using the guild ratio to characterize pelagic rotifer communities. Hydrobiologia, 662(1), 157-162.    Obertegger, U., & Manca, M. (2011). Response of rotifer functional groups to changing trophic state and crustacean community.)

Answer: Thank you for the comment. According to Jeong et al. (2014), there are a total of 85 species of crustaceans that occurred in South Korea. Of these, only two species of the Leptodera order were reported as carnivorous crustaceans, and the occurrence of the Bythotrephes order was not reported in South Korea. The Leptodora order was analyzed without excluding it because the frequency of occurrence was very low at 0.41 % during the study period.

Genus

RA(%)

Daphnia

50.92

Bosmina

32.93

Diaphanosoma

8.97

Ceriodaphnia

3.43

Chydorus

1.14

Bosminopsis

0.81

Scapholeberis

0.51

Alona

0.51

Leptodora

0.41

Moina

0.17

Picripleuroxus

0.08

Disparalona

0.04

Sida

0.03

Simocephalus

0.03

Monospilus

0.01

Q2. Lines 40-43: “The emergence of ….. order, which is a filter feeder [8-11]” Not all cladocera are filter feeders!

Answer: Thank you for the comment. As the reviewer suggested, we have changed the sentence in the revised manuscript as follows;

“The occurrence of the CWP is favored by an increase in fine phytoplankton itself and an increase in biomass of the of the Cladocera order, which is mostly a filter feeder” (Line 40 - 42).

Q3. Lines 45, 229, 260, 271, 276: Daphnia in italic

Answer: Thank you for the comment. As the reviewer suggested, we have changed to Daphnia in the revised manuscript.

Q4. Line 58: HRT, explain the meaning of the abbreviation

Answer: Thank you for the comment. As the reviewer suggested, we have explained the meaning of HRT (hydraulic retention time) (Line 60).

We have addressed all the reviewer’s comments and corrections in the text and would like to thank them for his/her assistance.

Round 2

Reviewer 2 Report

In line with my opinion expressed earlier, I believe that the manuscript presents interesting and valuable results of research into the relationship between many biological and environmental factors in the occurrence of the clean water phase in an artificial dam reservoir.
The significance of the manuscript is emphasized by the long period and frequency of data collection and the correct selection of research sites that allow for the analysis of various types (lake character, river character)  of particular fragments of the studied reservoir. This allowed to obtain interesting results as a result of statistical analyzes (mutual correlations), as well as the use of the Bayesian network.
All the important comments submitted during the review were taken into account by the Authors and the manuscript was changed and significantly improved, so I recommend it for printing.